# The Role of Information and Communication Technology (ICT) for Older Adults’ Decision-Making Related to Health, and Health and Social Care Services in Daily Life—A Scoping Review

**DOI:** 10.3390/ijerph19010151

**Published:** 2021-12-23

**Authors:** Susanna Nordin, Jodi Sturge, Maria Ayoub, Allyson Jones, Kevin McKee, Lena Dahlberg, Louise Meijering, Marie Elf

**Affiliations:** 1School of Health and Welfare, Dalarna University, 791 88 Falun, Sweden; may@du.se (M.A.); kmc@du.se (K.M.); ldh@du.se (L.D.); mel@du.se (M.E.); 2Population Research Center, Urban and Regional Studies Institute, Faculty of Spatial Sciences, University of Groningen, P.O. Box 800, 9700 AV Groningen, The Netherlands; j.l.sturge@rug.nl (J.S.); l.b.meijering@rug.nl (L.M.); 3Department of Physical Therapy, University of Alberta, Edmonton, AB T6G 2G4, Canada; cajones@ualberta.ca; 4Aging Research Center, Karolinska Institutet & Stockholm University, 171 77 Stockholm, Sweden

**Keywords:** autonomy, decision-making, health and social care services, older adults, participation, scoping review

## Abstract

Information and communication technology (ICT) can potentially support older adults in making decisions and increase their involvement in decision-making processes. Although the range of technical products has expanded in various areas of society, knowledge is lacking on the influence that ICT has on older adults’ decision-making in everyday situations. Based on the literature, we aimed to provide an overview of the role of ICT in home-dwelling older adults’ decision-making in relation to health, and health and social care services. A scoping review of articles published between 2010 and 2020 was undertaken by searching five electronic databases. Finally, 12 articles using qualitative, quantitative, and mixed-method designs were included. The articles were published in journals representing biology and medicine, nursing, informatics, and computer science. A majority of the articles were published in the last five years, and most articles came from European countries. The results are presented in three categories: (i) form and function of ICT for decision-making, (ii) perceived value and effect of ICT for decision-making, and (iii) factors influencing ICT use for decision-making. According to our findings, ICT for decision-making in relation to health, and health and social care services was more implicitly described than explicitly described, and we conclude that more research on this topic is needed. Future research should engage older adults and health professionals in developing technology based on their needs. Further, factors that influence older adults’ use of ICT should be evaluated to ensure that it is successfully integrated into their daily lives.

## 1. Introduction

Information and communication technology (ICT) can potentially support older adults in making decisions and increase their involvement in decision-making processes. This article presents an overview of the literature regarding the role of ICT for older adults living at home and their health-related decision-making in daily life. We used the term decision-making broadly; thus, it may include decisions in the person’s everyday life related to health and social care services as well as health-related decisions. In addition, there are various terms for ICT, such as eHealth, telehealth, and telemedicine, which are often used interchangeably [1]. In this review, we used ICT to describe devices and applications that facilitate digital information transmission [2]. 

Older adults can experience frail health in terms of both cognitive and physical impairments, functional limitations, and loss of autonomy [3] with the risk of participation restrictions regarding involvement in valued life events [4], including social, civic, and recreational activities [5]. Frail health in older adults can lead to a need for support from health and social care services to manage their daily lives and age well [6]. Older persons’ well-being is closely related to autonomy [7], and studies have shown that autonomy and participation can contribute to a sense of control [8], which plays an essential role in improving physical and cognitive functioning and increasing longevity [9]. In Western countries, there is a strong emphasis on promoting people’s autonomy and participation by providing care and services based on the needs of the individual [10], also described as a person-centred approach [11]. Many health and social care providers have expressed a commitment to providing this approach, and internationally, this is reflected in the World Health Organization’s global strategy for person-centred healthcare [12].

Most older adults want to live at home as long as possible [13,14]. Research has found that older adults living at home can maintain their physical function better and experience less depression than those living in institution [15,16]. Given that only those in the most frail health can access residential care facilities [17], older adults who live at home often require daily care and support [18]. 

Decision-making is complex and influenced by several factors, such as people’s health, habits, beliefs, social norms, and a desire to do the right thing [19]. Decisions for older adults who experience functional limitations can concern a broad range of daily life situations, such as home help (e.g., personal care, housekeeping, meals, transportation), adaptation of the home environment, healthcare decisions, or financial decisions [20]. Moreover, older adults can face even more complex decisions, such as the choice between staying in the home or moving to a care facility [21]. Despite an awareness of the risks of frail health with increasing age, research indicates that many community-dwelling older adults have difficulties making decisions on their future needs due to, for example, uncertainty about the future and sufficiently good present health [22]. Several actors are usually involved in decisions related to health and social care services for older adults, including their families and caregivers [19,23]. Different actors may see situations differently and emphasize different needs and preferences in the decision-making process [19,24]. Positive interactions between the older person, a supportive environment, caregivers, and the social network, such as family, friends, and neighbours, are a prerequisite for involvement in decision-making [23,25,26]. Furthermore, decision aids in the form of ICT can help older adults make health-related decisions [27]. 

Recently, there has been a focus on how ICT can potentially support older adults in staying at home despite functional declines [28,29]. Studies have shown that ICT support can empower older adults to be more engaged in decisions regarding health and social care services [30,31]. For instance, health applications on mobile phones and tablets have made it possible to monitor illnesses [32], and to communicate directly with healthcare professionals [33,34]. Additionally, ICT can increase self-management among people with chronic conditions living at home, as they participate in their own health concerns in a more effective way [32]. During the COVID-19 pandemic, the use of ICT has been brought to the forefront, and research has been conducted on vulnerable groups in society [35]. For example, Goodman-Casanova and colleagues showed that ICT consisting of a television-based integrated technology could promote active ageing of older adults with cognitive impairments living in their own homes via functions, such as health education and video calls [36].

In general, older adults’ experiences of and views on using ICT vary. While ICT can offer many benefits to older adults in need of care and support [37], for some people, ICT use implies the loss of valued personal contact; for example, ICT-based care can change the way care is delivered [38]. Therefore, when developing new technological solutions to support older adults’ participation in health and social care services to promote sustainable use of ICT over time, it is essential to integrate their perspectives and meet individual needs [39,40]. A large number of studies have been published on technology use and acceptance among older adults [41,42], and there is an increasing interest among older adults in using ICT [43]. However, people with frail health use ICT less often than younger people [44]. In addition, the frailty status among community-dwelling older adults is associated with ICT use where persons in frail health use ICT to a lesser extent compared to those with better health [45]. Although the range of technical products has expanded enormously in various areas of society with, for example, computerized devices and web-based applications, knowledge is lacking on the use of ICT for decision-making among older adults, especially with regard to daily life decisions. Thus, the overall aim of the study was to provide an overview, based on the literature, on the role of ICT in home-dwelling older adults’ decision-making in their daily life in relation to health, and health and social care services.

## 2. Materials and Methods

A scoping review methodology outlined by Arksey and O’Malley [46], enhanced by Levac, Colquhoun and O’Brien [47] was used and recommendations by the Joanna Briggs Institute were applied. This involved the following stages: (1) identifying the research question; (2) identifying relevant studies; (3) selecting studies; (4) charting the data; and (5) collating, summarizing, and reporting the results [46]. An optional sixth stage, consultation, was not applied in the present review. 

### 2.1. Identifying the Research Question

The following research questions guided our review:What types of ICT are used for decision-making in relation to health, and health and social care services from the perspective of older adults?In what ways is ICT used for decision-making in relation to health, and health and social care services from the perspective of older adults?Which factors can influence ICT use for decision-making in relation to health, and health and social care services from the perspective of older adults?

### 2.2. Identifying Relevant Studies

The keywords and search terms were developed in consultation with a specialist librarian and two researchers with expertise in the field of health-related decision-making. To identify as many relevant articles as possible, a broad search strategy was used. Keywords pertaining to ICT, older adults and decision-making were combined using the Boolean operators AND/OR. The search is presented in Table 1, including all synonyms. The searches were performed by two of the authors (SN and MA) in the following five databases: Cinahl, PsycInfo, PubMed, Scopus, and Web of Science. Reference lists of included articles were screened to identify further relevant articles. 

### 2.3. Selecting Studies 

The inclusion and exclusion criteria were based on the population, concept, and context (PCC) framework (the Joanna Briggs Institute, 2015). In short, articles were selected if they contained first-hand experiences from people aged 65 years or older (population) and described ICT for daily life decisions in relation to health, and health and social care services (concept) used in the home environment (context). In addition, the following inclusion criteria were applied: (i) empirical studies (qualitative, quantitative, or mixed-methods) published as articles in peer-reviewed, scientific journals; (ii) articles written in English; and (iii) articles published from 2010 to 2020. Articles that did not include empirical data (e.g., editorials, expert opinions, grey literature) were excluded. See Table 2 for inclusion and exclusion criteria.

A systematic study selection process was applied following the recommendations in the Preferred Reporting Items for Systematic Reviews and Meta-Analysis extension for Scoping Reviews (PRISMA-ScR) [48]. The study selection was conducted in a two-step process. First, titles and abstracts for all hits were independently screened by two reviewers (SN and MA) and either included or excluded for further review. Second, the full texts of all potentially relevant articles were independently reviewed by the same two reviewers. Five of the articles were also screened by another author (JS) to confirm that they corresponded to the research questions [47]. 

### 2.4. Charting the Data 

The data were extracted by two investigators (SN, MA) using a developed standardized data extraction template describing the following: authors, publication year, country of origin, study aim, design and methods, participants, ICT characteristics and main findings (see Table 3). Regarding articles that involved a mix of study participants (e.g., staff, older adults), only the data on first-hand experiences from older adults were included in the analysis. Quality appraisal of each study was not performed [46]. 

### 2.5. Analysing the Data

As recommended by Peters and colleagues, a basic descriptive analysis was conducted to reflect our study aim [49]. In our study, the nature of decision-making in relation to ICT was summarized narratively. The descriptive analysis was conducted initially by three of the authors (SN, MA, JS), and discussed and revised in the research group. The following steps were conducted:
All included articles were read several times to identify relevant content;Relevant content was grouped into three broad categories corresponding to the research questions and refined into eight subcategories.

### 2.6. Ethical Considerations

As secondary data analysis, ethics approval was not required for this scoping review.

**Table 3 ijerph-19-00151-t003:** Summary of included articles (n = 12).

Authors (Year) Country	Study Aim	Design and Methods	Participants (Sample Size), Age	ICT Characteristics	Main Findings
Algilani, Langius-Eklöf, Kihlgren, Blomberg (2016), Sweden [50].	To develop and test feasibility and acceptability of an interactive ICT platform integrated in a tablet for collecting and managing patient-reported concerns of older adults in home care.	Mixed-method design Interviews Logged quantitative data	Older adults (n = 8), 67–90 years old Nurses (n = 3)	ICT platform for assessment of health and wellbeing, healthcare advice and links to websites for information, risk assessment model for alerts, connection to a monitoring web interface, graphs to view reported health concerns.	Via the platform, self-care advice was regarded by the participants as a good asset when they needed information on health issues and what they could do themselves. It enabled communication between them and the nurses, and could facilitate participation. The platform had relevant content and was perceived to be easy to use although technical challenges were identified (e.g., issues related to font size and logging in).
Demiris, Thompson, Boquet, Shomir, Chaudhuri, Chung (2013), USA [51].	To evaluate the perceived usability and effectiveness of a telehealth wellness kiosk in an independent retirement community as well as privacy considerations.	Qualitative design Focus groups	Older adults (n = 12), 65 years or older	A telehealth kiosk for assessment of physiological parameters, online questionnaires, a library of educational videos, and a brain fitness web-based software solution.	The participants appreciated the cognitive assessments and to frequently capture physiological parameters. Printouts of the data made it possible to share data with family and clinicians. The older adults valued health-related decision-making and saw the telehealth kiosk as a tool to improve independence and control over their health status. Technical challenges were identified (e.g., problems to handle computer mouse).
Demiris, Thompson, Reeder, Wilamowska, Zaslavsky (2013), USA [52].	To demonstrate how informatics applications can support the assessment and visualization of older adults’ wellness.	Mixed method design Measurements Questionnaires Focus groups	Older adults (n = 27), 78–94 years old	A platform that integrates three components; a software application capturing functional parameters, a telehealth kiosk, a software application assessing cognitive parameters.	The participants had positive experiences of the ability to capture vital signs and transmit via Bluetooth, and the possibilities with a personal journal for sharing information with family and caregivers. Engagement in cognitive performance activities was appreciated, and the opportunity to socialize and interact with others in the community. Technical support from staff and diverse tools for user preferences were needed.
Dupuy, Consel, Sauzeon (2016), France [53].	To promote self determination-based theory into the design of gerontechnologies.	Quantitative design Platform development and evaluation Questionnaires	Older adults (n = 34), 82 years old on average	An assisted living platform for applications that utilize a range of devices (e.g., motion detectors, contact sensors, smart switches) and software components (e.g., calendar, photo album, address book).	The use of the platform improved self-determination performance; autonomy, self-regulation, empowerment, and self-realization of the participants. The platform could enable the user to make decisions about assistance needed to live autonomously and conduct meaningful activities. Easy to use technology.
Göransson, Wengström, Ziegert, Langius-Eklöf, Blomberg (2020), Sweden [54].	To describe and evaluate the experiences of self-care support and sense of security among older persons using an interactive app to report health concerns.	Mixed method design Measurements Questionnaires Interviews	Older adults (n = 17), 70–101 years old	An interactive ICT-platform for application in smartphones or tablets, and used in the assessment of health concerns and self-care support among older persons with home care. Included access to self-care advice, graphs and risk assessment sending alerts to nurses.	The platform was used for self-care advice in different ways by the participants, and was perceived to provide accurate information. The self-care advice could be a trigger to search for more health-related knowledge. It was beneficial to access advice directly without the need to contact healthcare staff. The platform was perceived as a way to interact and communicate with the healthcare staff.
Göransson, Eriksson, Ziegert, Wengström, Langius-Eklöf, Brovall, Kihlgren, Blomberg (2018), Sweden [55].	To explore the experiences of using an app among older people with home-based health care and their nurses.	Qualitative design Interviews Focus groups	Older adults (n = 17), 70–101 years old. Nurses (n = 12)	An interactive ICT-platform for application in smartphones or tablets, and used in the assessment of health concerns and self-care support among older persons with home care. Included access to self-care advice, graphs and risk assessment sending alerts to nurses.	Via the platform, the participants were stimulated to learn about their health concerns. Self-care advice increased their ability to care for themselves and supported self-confidence. Also, it enhanced communication and enabled participants to report health concerns more precisely. Their self-confidence increased as they were able to use the technology. Technical challenges were identified (e.g., issues related to font size on buttons and logging in).
Harrefors, Axelsson, Sävenstedt (2010), Sweden [56].	To describe healthy older couples’ perceptions of using assistive technology services when needing assistance with care.	Qualitative design Interviews	Older adults (n = 23), 70–83 years old.	Technology services from technical aids for daily living to IT-based services for security, communication and remote consultation	Regular health monitoring made the participants feeling more secure at home. Technology could assist and support older persons in frail health, and was a way to communicate with nursing staff and friends. Also, it was perceived to provide an opportunity to live at home for a longer time.
Irizarry, Shoemake, Lee Nilsen, Czaja, Beach, DeVito Dabbs (2017), USA [57].	To explore attitudes toward portal adoption and its perceived usefulness as a tool for health care engagement among older adults with varying levels of health literacy and degrees of prior patient portal use.	Mixed method design Phone survey Focus groups	Older adults (n = 100), 65–97 years old.	Patient portals in general for access to personal health information and patient-provider communication. Examples of common portal features are health information, medication management, health results, and communication and appointment setting.	Overall, the participants reported that it was valuable to have all their personal medical information and clinician contact information in one place. As for health information in the portal, the experiences of the participants varied (e.g., some lacked individualized information). Outdated or incorrect medical data in the portal led to frustration among older adults. There was a need for training and support to manage technology.
Dickman Portz, Bayliss, Bull, Boxer, Bekelman, Gleason, Czaja (2019), USA [58].	To use the technology acceptance model as a framework for qualitatively describing the user interphase and experience, intent to use, and use behaviors among older patients with multiple chronic conditions.	Qualitative design Focus groups	Older adults (n = 24), 65 years and older	A patient portal providing personal health information related to patient diagnosis, prescriptions, laboratory results, vaccination records. Health management features that are designed to foster healthy eating and exercise habits incorporate personalized assessments and self-management health tools.	The participants thought that the portal was useful for get access to health information and addressing health concerns without a clinic visit, especially for those living in rural areas. The portal was seen as useful for communicating with healthcare providers. Participants felt confident when they managed to use the portal. Although the portal was perceived as easy to use, technical challenges were also identifed (e.g., issues related to font size or logging in).
Robben, Perry, Huisjes, van Nieuwenhuijzen, Schers, van Weel, Olde Rikkert, van Achterberg, Heinen, Melis (2012), The Netherlands [59].	To establish the outcomes of the implementation process of the Health and Welfare Information Portal, which implementation strategies and barriers and facilitators contributed to these outcomes, and how its future implementation could be improved.	Mixed method design Survey Interviews	Older adults (n = 290), 70 years and older Professionals (GPS:s, nurses, gerontological social workers, other) (n = 169)	A personal, Internet-based conference table for multidisciplinary communication and information exchange for frail older people, their informal caregivers, and professionals. The table is considered to be both a shared electronic health record and personal health record.	Via the portal, the participants could keep control over their own care. The older adults’ messages were quickly answered by their own general practitioner. It enhanced participation of informal caregivers and general practitioners and facilitated the involvement of older adults. Although the portal was perceived as user-friendly, barriers for the older adults were found such as not being comfortable with using computers or not being familiar with the portal.
Schmidt, Behrens, Lautenschlaeger Gaertner, Luderer (2019), Germany [60].	To gain a better understanding of how care and case management (CCM) in general is perceived by older people (65+) living alone and what they think about the CCM monitoring process used during video conferences.	Mixed method design Interviews Measurements	Older adults (n = 40), 64–92 years old (inclusion criteria was 65 years or older)	Video conferences via tablet PCs enabling information and advice from nurses and social workers, and communication between older adults. Possibility to download further information material.	Video conferencing was perceived by the participants as valuable for social contact and communication. People with reduced mobility found it useful to have access to case managers and other healthcare professionals for support and advice (e.g., carers, physiotherapists, social workers, dentists). It compensated for isolation and enabled independent participation. The main problem with technology was poor internet connection in rural areas. Also, need for touch function training was identified.
Willard, Cremers, Man, van Rossum, Spreeuwenberg, de Witte (2018), The Netherlands [61].	To support frail older adults in their independence and functioning, by stimulating self-care and providing reliable information, products and services.	Mixed method design Observations Interviews Measurements	Older adults (n = 33) 65 years and older	Online community care platform containing 11 functions; emergency call, services, contacts, clock, calendar, medication reminder, news, sending and receiving messages, information about the community, information from municipalities, and games.	The participants mainly used the platform functions of contacts, services and messaging. The platform was regarded to contribute to the social participation, the self-management competencies, and with their social cohesion in the community. However, only a minority thought that the platform had added value to them. Although the participants perceived the platform easy to use there were some technical challenges (e.g., issues related to logging in, and function arrangement).

## 3. Results

The search resulted in a total of 2308 articles. After the exclusion of duplicates, 1651 titles and abstracts were screened. The articles that did not meet the inclusion criteria were removed. All remaining articles were read in full (n = 32), and articles were removed if they did not meet the criteria (n = 22). Additional articles were identified from the reference lists of eligible articles (n = 2). In total, there were 12 relevant articles. The search process is presented in the PRISMA flow chart (see Figure 1).

### 3.1. Study Characteristics

The included articles are described in Table 3 [50,51,52,53,54,55,56,57,58,59,60,61]. Four of these articles reported the same two research studies but focused on different aspects of these studies. Most of the articles concerned ICT in relation to healthcare, such as communicating with healthcare professionals and receiving advice on health issues. The articles were published between 2010 and 2020 in journals representing biology and medicine, nursing, informatics, and computer science. There were eight articles from European countries and four from the United States. The number of older adults (65+ years of age) in the included articles ranged from eight [50] to 290 [59]. Seven of the articles were of mixed-method design [50,52,54,57,59,60,61], four articles used a qualitative design [51,55,56,58], and one used quantitative design [53].

### 3.2. Summary of Key Findings

The findings are presented in three main categories with eight subcategories. The main categories were constructed based on the three research questions: form and function of ICT for decision-making; perceived value and effect of ICT for decision-making; and factors influencing ICT use for decision-making (see Figure 2).

#### 3.2.1. Form and Function of ICT for Decision-Making

There was a broad range of ICTs that are presented below in the following three subcategories: online platforms, web-based portals, and other ICT-based services.

##### Online Platforms

Online platforms can be used to support older adults’ decision-making processes. In several studies, online platforms were used with integrated features, such as information websites, sending and receiving messages, and health assessments [50,52,53,54,55]. For example, the use of an interactive platform for applications in smartphones or tablets was described in three studies [50,54,55]. The platform included assessing older adults’ health problems for direct transfer to care staff, access to self-care counselling, risk assessment, connection to a web interface, and diagrams of reported health problems [54,55]. Another study reported a platform for community services such as personal alarms, video call services, information on local events and activities, care consultations, reminders of medications, and health advice [61].

##### Web-Based Portals 

In three studies, the older adults used web-based patient portals to access personal health information and communicate with healthcare professionals [57,58,59]. The portal developed by Robben and colleagues consists of personal health information, a messaging system for the older adults and staff, and education materials. Because the portal enabled access for all involved, they could stay informed about the older adult’s situation, and everyone could share information [59]. The portal used in the study by Portz et al. [58] also included the ability to store personal health information, a communication system between older adults and caregivers, and features to promote healthy eating and exercise habits.

##### Other ICT-Based Services

Finally, two studies used other ICT-based services [56,60]. In the study by Schmidt et al., video conferences via tablet computers were used in which older adults could receive information and advice from healthcare professionals. Additionally, older adults could have contact with people in similar life situations and with similar interests [60].

#### 3.2.2. Perceived Value and Effect of ICT for Decision-Making

The category perceived value and effect of ICT for decision-making was structured into three subcategories: assessing and sharing information; communicating and interacting with others; and being more independent and secure.

##### Accessing and Sharing Information

Most articles described the value of ICT for decision-making in terms of accessing and sharing information that actively involved older adults in their health and social care services [50,51,52,54,55,56,57,58,59,60,61]. For instance, self-care advice via ICT was regarded by older adults as valuable when they needed knowledge and information on health issues and what they could do to have control over their own care without having to visit their healthcare facility [50,54,55,60]. The study by Schmidt et al. [60] found that video conferences had many everyday benefits for older adults by providing them with opportunities to make enquiries based on particular life situations and by involving different healthcare professionals in providing valuable information and contacts. Furthermore, some studies reported on ICT for sharing information with family members and other informal caregivers [51,52,59,60]. For example, Demiris and colleagues emphasized the importance of a holistic perspective where older adults themselves, their informal caregivers and healthcare professionals had access to up-to-date information on matters relating to health and well-being, which in turn could support decision-making about the older person’s life situation [52]. 

##### Communicating and Interacting with Others

ICT was related to decision-making via communication and interactions that occurred. According to most articles, ICT could enhance older adults’ communication and interaction with professionals and others [50,51,52,54,55,56,58,59,60,61]. Older adults viewed ICT as useful for engaging with their healthcare professionals [55,56]. For instance, using an app was described by the older adults as a new way to communicate with nurses enabling them to report their health concerns in more detail [54,55]. Additionally, interactions with healthcare professionals via ICT were regarded as especially valuable by persons living in rural areas [58]. Communicating via ICT for social contacts was also described as an important aspect, not least for people with health issues [55,56,61] and reduced mobility [60]. In the study by Willard et al., older adults preferred to use the examined platforms’ video call service function, which could support social interactions with family and friends as part of the decision-making process with regard to health and social care services [61].

##### Being More Independent and Secure 

Our findings indicated that ICT could relate to decision-making in terms of independence, security, control, and self-confidence [51,53,55,56,61]. Information on personal health and advice from healthcare professionals via ICT gave older adults the confidence to manage their own health issues at home [53,54,55], which was regarded especially important for those living alone [54,55]. Regular and quick communication with healthcare professionals via ICT enabled older adults to feel secure and confident [55,56]. Additionally, older adults became more active in their own care as they were stimulated to learn more about ICT and their health issues [54,55]. In addition, some studies have shown that ICT use itself could contribute to older adults’ self-confidence [53,58]. For instance, older adults felt proud when managing an ICT application without help from others [50,55].

#### 3.2.3. Factors Influencing ICT Use for Decision-Making

Factors reported to influence ICT usability are presented in two subcategories: relevant content based on user needs and training, and support and technical issues.

##### Relevant Content Based on User Needs

Several studies found that relevant and up-to-date content facilitated the use of ICT for decision-making [50,54,55,57,61]. For instance, one study demonstrated that topics that are relevant to older persons’ health and daily life need to be covered without being too general [50]. Irizarry et al. found that outdated or incorrect health data resulted in a sense of frustration among older people [57]. Other studies have indicated that older adults differ in their experience of ICT use and health issues and thus require individual adaptations [50,51,55,61]. For instance, one study explored the value of providing multiple technical options for an older adult to choose from [52]. The findings also showed that text and buttons must be designed to be visually clear [51,54,55].

##### Training, Support and Technical Issues

Limited previous computer experience was found to be a barrier to ICT use [57,59], and the benefits of training and support were raised [52,58,59]. For instance, Portz et al. suggested that older adults be provided with specific training on how to use a portal and its features [58], while Demiris et al. emphasized that older adults with particular health problems, such as tremor and hearing loss, require customized training to operate the ICT [52]. Training for families and caregivers was also recommended so that they could in turn support older adults [52]. Some studies reported technical issues, such as logging in, locking the tablet or charging the device [50,55,58,61]. For instance, older adults forgot their username and password, or became anxious and frustrated, which made logging in problematic [55,61]. A functioning network connection is a prerequisite for using online ICT [57,60], and poor internet connection was especially problematic in rural areas [60].

## 4. Discussion

This review is based on studies involving older adults. Despite the increasing emphasis on guidelines and policies to strengthen autonomy and participation in vulnerable groups, studies on how ICT can support older adults in decision-making with regard to health and to health and social care services seem scarce. According to our results, ICT for decision-making is vaguely described but could be related to having access to information, communicating with relatives, friends, and staff, and supporting independence. Different ICT devices were used for these reasons. 

The findings identified three main types of ICT to support decisions: online platforms, web-based portals, and other ICT-based services. Several of the platforms and portals had integrated features such as personal health information, message exchange, and health assessments. These ICT devices can be essential tools for older adults to be able to make decisions in their daily life. Integrating a wide range of features into a single digital solution has been proposed to prevent fragmentation [62]. This may be particularly valuable to older adults with frail health who require different levels of care and support and need to contact health and social care services. In that respect, our review may be interesting as several articles indicated that ICT devices with integrated functions can facilitate access to information compared to having to search in several different places. However, identifying a suitable ICT device can be a challenge, as it is not always clear what services are available and what is the focus on a particular digital solution [62]. Moreover, it is well recognized that older adults do not use ICT devices on a large scale and that it is challenging to successfully introduce such devices for this group [63]. Hence, this must be addressed in the health and social care sector, as it cannot be taken for granted that technology solutions that are introduced are also used.

The main value of ICT for decision-making seems to be the ability to access and share information that actively involves people in their health and social care services. ICT can provide older adults with relevant information as a basis for deciding on health and social care services and, to some extent, compensate for poor information provided in other parts of the care system This is in line with other studies reporting that relevant and accessible information is crucial for decision-making [64], especially for older adults with frail health [65]. Examples of relevant information that may be of value are different types of health and social care services that are available. Turnpenny and colleagues found in their systematic review that people with frail health are not always aware of appropriate options for their services [26]. It has been shown that older adults learn about other alternatives by chance or through previous contacts with care and service. Additionally, older adults accept proposed alternatives without the possibility of considering other alternatives [65]. This can intrude into older adults’ autonomy and right to make informed decisions and may lead to an imbalance of power between older adults and societal services. In Sweden, both the Social Services Act and the Patient Act emphasize the importance of adequate information for being able to participate in decisions related to the support needed. Today, ICT is more important than ever since most societal information is ICT based. Thus, it was striking that so few articles focused on technology in relation to decision-making, and there seems to be a need for more knowledge on how ICT can be used to guide decision-making. 

Another key finding in our review was that ICT could relate to decision-making in terms of independence, security, control, and self-confidence among older adults. For instance, receiving information and advice via ICT seemed to strengthen older persons’ abilities to manage their own health issues. Being dependent on support can be a challenge to maintain autonomy, and many older adults feel that they have less influence on everyday life when their dependence on support from others increases [7]. Previous studies have described the widespread use of ICT to improve independence and contribute to the autonomy and participation of older adults in different life situations [66]. There are also indications that older adults who use technology regularly have higher self-confidence than those who are less frequent users of technology [67]. This links to person-centredness, which is about placing the person at the centre of his or her life and being involved in decisions that promote autonomy and control over life situations [11]. According to several studies, the ability of older persons to make relatively small daily life decisions has a significant impact on their sense of control [68,69,70]. ICT has been proposed to support a person-centred approach by encouraging people to take an active role in their own health and decision-making process [71]. Therefore, ICT use can greatly benefit older adults in terms of making them in charge of their own life situations and enabling participation in decisions affecting them in their daily lives.

Despite the potential benefits of ICT use for decision-making, there are some obstacles. For many older adults, ICT can be overwhelming and perceived as challenging to learn and navigate [72]. For instance, our results showed that limited previous computer experience could be a barrier to ICT use. Studies have shown that relatives often play an essential role in ICT use among older adults, such as performing internet searches on behalf of their older relatives [65]. Positive experiences may occur when older adults collaborate and socially interact with others by means of a concreate task to solve. Moreover, valuable social contacts can be created through generations when younger people support their older relatives. However, it can be problematic not to be able to handle new technology when important decisions have to be made. One way to support older adults in their ICT skills can be by introducing courses led by other older adults with more experience using technology. For instance, in Sweden, a nonprofit, independent association called SeniorNet helps older adults take part in the opportunities that digitalization offers in society (https://seniornet.se/, accessed on 14 November 2021). In the Netherlands, there is a similar association, the SeniorWeb.NL, which runs a community platform for older adults (www.seniorweb.nl, accessed on 15 December 2021).

Other actors include staff, who play an important role in encouraging older adults to use technical devices. This requires that the staff themselves are familiar with technology and have experiences and interests in using it [73]. According to a recent study staff in municipal care of older people had positive attitudes towards new technology in general. However, those staff members who had closest contact with older adults in their daily work had little influence on the technical devices being procured [74]. This can lead to resistance to technology use among both staff and older adults. Hence, to successfully deploy ICT into health and social care services, staff members need to be involved in the process, and by starting “on the floor”, the implementation of technology can occur more quickly and result in smarter solutions [73]. Furthermore, it has been suggested that older adults need to begin using technology before they become too frail and dependent on care and support. If older adults become familiar and confident in using ICT earlier in life, there is much to gain, not only for the older persons themselves but also for the entire health and social care support system.

This review shows that for ICT to be used, its content and design must be adapted to older adults, and several of the included articles reported frustration among users when the content did not reflect their needs. This is consistent with other studies emphasizing that technologies must address the personal, social, and physical contexts of older adults [42]. If older adults do not find a value for using ICT in their daily lives, the thresholds become too high, which may hinder the possibilities for support in decision-making. It is not only the content but also the design of the devices that must fit with the functional status of the older population. For example, buttons that are clear and easy to use and devices that are straightforward to navigate can facilitate ICT use. To overcome barriers and offer technology tailored to user needs, older adults themselves have to be involved in the process of creating new ICT solutions. This means that technology companies need to invite older adults to participate in the entire development process. There are promising projects in which technology solutions have been designed in co-design processes involving different stakeholders. For instance, an international project applied the double diamond method involving the stages of generating ideas, modelling a prototype, and testing together with older adults, caregivers, representatives from technology companies and professionals from the health and social care service [75]. 

### Methodological Strengths and Limitations

A scoping review was considered an appropriate method to explore the range of existing literature on this topic. Given the rapid development in technology, it is common to limit searches to a 10-year period in literature reviews in this field [76], and we included articles published from 2010 to 2020. Although we followed the recommended methodology and conducted a systematic search strategy, it is possible that the search strategy and inclusion/exclusion criteria did not identify all relevant articles. For instance, we might have missed relevant articles due to the restriction to English language publications only or not including grey literature. As per the scoping review framework, no quality assessment has been made of each article, this may have affected the results (e.g., studies using small sample size, non-randomized sample, lack of control groups). Additionally, decision-making is a multifaceted concept used relatively recently in the literature. It is closely related to concepts, such as participation, choice and control and might be difficult to operationalize in practice. Still, the search terms used were broad in order to address this potential problem. Another limitation is that certain articles were based on the same research studies, which may have placed too much emphasis on specific results. However, these articles focused on different aspects of the research and therefore made unique contributions to the review. Most articles came from European countries, which can be seen as a limitation as the results may not be applicable to countries with other health and social services systems. Finally, a majority of the articles were published in the last five years, indicating a growing interest in the field. 

## 5. Conclusions

This review provides an overview of the existing literature on the role of ICT in home-dwelling older adults’ decision-making in daily life related to health, and health and social care services. Despite the increasing number of publications on ICT to support an ageing population, there seems to be a need for further research to increase our understanding of how ICT can be used to guide older adults’ decision-making. According to our findings, ICT for decision-making was more implicitly described than explicitly described, for instance, in terms of prerequisites for making decisions such as having access to information and communicating with staff and others. Additionally, the incorporation of several functions into one solution through ICT seemed to facilitate ICT use for older adults and thereby contribute to decision-making. ICT holds potential to make life better for older adults as frailty increase and to support them in their health- and care-related decision-making processes. However, older adults are not a homogeneous group; rather, they are individuals with different needs, prerequisites, and interests. Thus, there is no one-size-fits-all digital solution. Future research should engage older adults and health professionals in developing the form and function of technology based on their needs. Further, factors that influence the adaption and willingness to use ICT should be evaluated to ensure that technology is successfully integrated into the daily lives of older persons.

## Figures and Tables

**Figure 1 ijerph-19-00151-f001:**
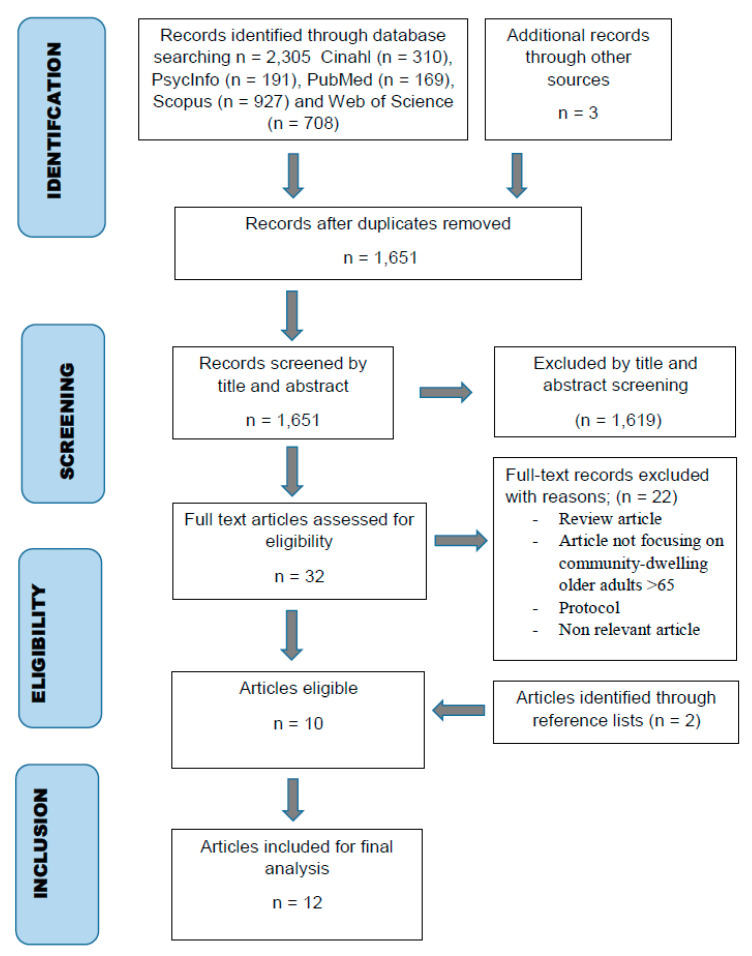
PRISMA flow chart.

**Figure 2 ijerph-19-00151-f002:**
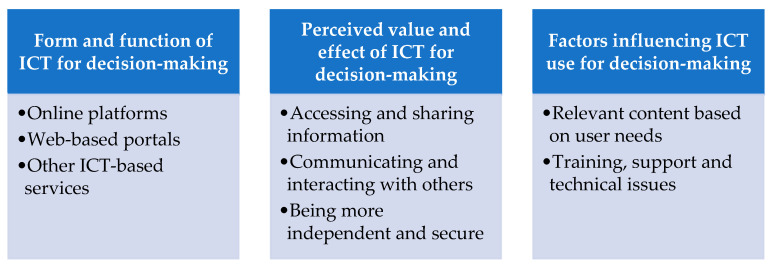
Categories and sub-categories.

**Table 1 ijerph-19-00151-t001:** Electronic database search strategy.

Search Terms Group A	Search Terms Group B	Search Terms Group C
Information and communication technology	older people	decision making
ICT	older adult	decision-making
platform	older person	user participation
internet	senior	user involvement
online	elder *	user preference

* This symbol enables unlimited searches for various word endings.

**Table 2 ijerph-19-00151-t002:** Inclusion and exclusion criteria based on the PCC framework.

PCC Framework	Inclusion Criteria	Exclusion Criteria
Population	Older adults aged 65 or older	Focus on other perspectives than those of older adults themselves
Concept	ICT for decision-making related to health, and health and social care services	Technical devices, such as sensors and alarms or robot technology
Context	Home environment	Focus on institutional care, such as residential care facilities

## Data Availability

The data that support the findings of this review are available from the corresponding author upon request.

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
