# Peer review of "The Role of Information and Communication Technology (ICT) for Older Adults’ Decision-Making Related to Health, and Health and Social Care Services in Daily Life—A Scoping Review"

_ijerph, 2021, doi:10.3390/ijerph19010151_

Round 1

Reviewer 1 Report

This is a very interesting publication on the role of ICT for older adults’ decision making related to health and use of services. The authors selected 12 studies on the topic that included qualitative data from seniors, as a source of ‘first hand experiences’. The introduction and discussion section describe interesting and relevant literature, including many reviews, other than their selection. The publication would gain readability if the position of their own selected 12 publications is made more clear throughout the text (see comments below). Also given their extensive reading, I would like more directions for future research. On the whole, this makes it a very interesting publication to read.

Abstract

  • In the abstract I miss information on results of their own review, please add a few sentences with results (and less on methods).

Introduction

  • Please end the introduction with the aim of this study. The introduction has much (relevant) information, and for me the sentence at the end ‘knowledge is lacking on the role…’ seemed rather contradictory to this. What specific knowledge was lacking, perhaps describe in a few more sentences.

Methods

  • In the research questions, please add that you want (only) information or opinions of seniors themselves. Please change ‘health and social care services’ to ‘ health, and health and social care services’.
  • Search terms: I had expected also studies on apps or technical devices (e.g. medication dispensers, used in daily life) I am not quite sure, did the search terms prevented these studies from emerging, or does this ICT not fall within the research question.
  • It is unclear to me why the exclusion criteria for concept (technical devices ..) was used, please explain.
  • Regarding paragraph 2.4 quality appraisal of each study: I am not interested that this ‘is not mandatory procedure’, but why the authors themselves regarded this as not relevant. Please add.

Results

  • I started reading table 3 first, before reading the results section. And this was quite confusing. The main text is well written. But the description of the studies in table 3 often was not clear to me, without the explanations in the text. In addition, I would like to know, did the seniors actually use the ICT in all studies?
  • Details on table 3
    • Please add the numbers of the references, as a link to descriptions in main text
    • Perhaps (sub) categories could be included, as a link to descriptions in main text?
    • Algilani 2016: what is ‘the elder care setting’ ? Dickman 2019, main findings: unclear, relation with decision making? Irizarry 2017: text under aim is method? Schmidt 2019: what is CCM?

Discussion

  • Please add in first paragraph, that selection extracted studies that involved seniors.
  • Qualitative studies give information on ‘perceived’ value and effect, as indeed is stated in the second category in figure 2. Sometimes sentences read, as if effects are already evidence based. E.g. line 384 ‘our results are promising’
  • Minor, line 442: NL also has it’s www.seniorweb.nl

Conclusion

  • In the last sentence, the authors conclude that it is key to involve older persons themselves in developing ICT. I would say, this is not the conclusion of the present paper, but the starting point of their study. Moreover, on several occasions they discuss that it is important to include healthcare professionals as well (line 445 encouraging role; line 452 staff members; line 471 co-design).
  • Line 501: ‘there seems to be a need for further research’. But what then exactly? There is so much information in this paper, that I cannot think of it. I would like more explanation or focus on this point. Perhaps an overview (table) of existing reviews and themes would also help.

Author Response

Please see the attachment "Responses to Reviewer 1 the 15th of Dec 2021"

Reviewer 2 Report

In line 101 you state that "However, people with frail health use ICT less often than younger 101 people [44]. Is tehre an indiciation in the use of ICT within the group of elderly homedwelling that elderly persons in frail health use ICT less often compared to other elderly home dwelling elderly?

PRISMA flow chart: on pdf an arrow is misplaced

Author Response

Please see the attachment "Responses to Reviewer 2 the 15th of Dec 2021"
